# Mellow Babies: A Randomised Feasibility Trial of an Intervention to Improve the Quality of Parent–Infant Interactions and Parental Mental Wellbeing

**DOI:** 10.3390/children11050510

**Published:** 2024-04-24

**Authors:** Lucy Thompson, Philip Wilson

**Affiliations:** 1Institute of Applied Health Sciences, University of Aberdeen, Aberdeen AB24 3FX, UK; lucy.thompson@abdn.ac.uk; 2Section of General Practice, Department of Public Health, University of Copenhagen, 1165 Copenhagen, Denmark

**Keywords:** feasibility trial, parent support, parent–child interaction, perinatal mental health

## Abstract

Mellow Babies aims to improve mothers’ mental wellbeing and the quality of their interactions with their baby. The feasibility of a definitive trial of Mellow Babies was assessed using a waiting-list randomised pilot trial (Clinicaltrials.gov: NCT02277301). Mothers with substantial health/social care needs and a child aged <13 months were randomly allocated either to a 14-week Mellow Babies programme or to receive usual care whilst on a waiting list for the intervention. Rates of recruitment and retention as well as participants’ views of their experience in this study were recorded. Outcomes were parenting behaviour, assessed by the blind-rated Mellow Parenting Observation System (primary) and self-report maternal wellbeing pre- and post-intervention/waiting period. We recruited 38 eligible participants: 36 (95%; 18 intervention, 18 control) completed baseline measures, and 28 (74%; 15 intervention, 13 control) provided post-intervention data. Two practitioners took part in feedback interviews. Intervention participants had significantly more positive interactions with their babies at post-intervention compared to those in the control group (*p* = 0.019), adjusted for pre-intervention scores. There was no significant improvement in mothers’ mental wellbeing on any measure. A definitive trial of Mellow Babies is feasible and should include longer follow up of mothers and the opportunity for fathers to take part.

## 1. Introduction

Psychologically informed very early intervention programmes across the ante- and postnatal period have shown promise in delivering cost-effective positive outcomes [1]. There is robust support for the view that early intervention to prevent and manage difficulties in the parent–infant relationship may produce benefits in improved parental psychosocial wellbeing, with a supporting impact on child development outcomes, such as attachment, language and social development, and a reduction in externalising and internalising behaviours [2,3,4]. O’Connor et al. [5] demonstrated that parental sensitivity in the early years has an enduring effect on attachment representation into adolescence, regardless of current parenting relationship quality. In their ground-breaking longitudinal work, Olds and colleagues [6,7,8] showed promising long-term impact in a reduction in rates of criminality, substance abuse, educational underachievement, chronic disease, and all-cause mortality in offspring from families who had taken part in an early intervention programme, compared to no treatment.

Group-based parenting interventions are a popular option and can be effective in improving outcomes for children [3], but more robust research is needed, specifically involving large, randomised groups and long-term follow-up. A seminal meta-analysis of sensitivity and attachment interventions in early childhood has shown that the most effective interventions use a moderate number of sessions (*n* = 5–16) and are focused on parental sensitivity [9]. There are a number of small trials of parenting interventions with young preschool children reporting impact upon parental sensitivity and attachment [10,11,12,13] and a number of useful systematic reviews and meta-analyses in the field [9,14,15,16,17,18]. Barlow and colleagues’ 2012 updated Cochrane review of postnatal parenting interventions designed to improve child emotional and behavioural adjustment [19] identified no eligible trials with families whose children had a mean age of less than two years, though a more recent review, also led by Barlow, of parent–infant psychotherapy [17] presented equivocal findings. Pontoppidan et al. (2016) also found equivocal results for universal interventions in infancy [20]. Kohlhoff and Cibralic (2022) showed evidence in their review of an impact of early-years (age 1–5 years) interventions on children’s externalising behaviour, with some evidence of sustained change at follow-up [21]. In their 2016 systematic review and meta-analysis, Barlow et al. concluded there was tentative support for the use of group-based parenting interventions in the under threes, with the evidence being limited by the lack of long-term follow-up data [3]. Goyder et al. (2022) confirmed this gap in the evidence remains, and called for more trial-level research into programmes which have already been developed and successfully implemented [1].

Mellow Babies is one such programme. It is a group-based intervention drawing on attachment theory and social learning theory designed for mothers with substantial problems in their relationships with their infants [22]. Typically, participating mothers have mental health difficulties, drug use problems, learning difficulties, forensic issues, or children already looked after on a statutory basis. The intervention consists of 14 whole-day sessions with a joint focus on maternal wellbeing and the parent–infant interaction. Transport and crèche facilities are provided and are integral to the intervention.

Although Mellow Parenting interventions (Mellow Babies is probably the best-evidenced among the Mellow Parenting suite) are quite widely used in a number of countries, no definitive randomised trials have been carried out. One small waiting-list controlled trial (*n* = 17) has shown improvement in mothers’ mood and quality of parent–infant interaction compared to the control [22]. Our meta-analysis [23] found Mellow Parenting programmes showed medium effects on maternal wellbeing and child behaviour problems, but there was a degree of heterogeneity and methodological weakness in the included studies. To date, there has been a single small waiting-list trial of Mellow Babies suggesting improvement in maternal mood and mother–child interaction [22], and a large pre–post evaluation (*n* = 91) showed completion of the intervention to be associated with significant improvements in anxiety and overall wellbeing, parenting confidence, and perceived closeness of the parent–child relationship (although parenting confidence was not significant in intention-to-treat analysis) [24]. A subsample of the same cohort took part in qualitative interviews, where thematic analysis characterised positive parental perceptions of Mellow Babies, including the opportunity to reshape interpersonal interactions [25].

We aimed to conduct a feasibility study to inform the design of a definitive RCT for Mellow Babies. Our primary purpose was to ascertain the rates of recruitment and retention in the trial and to gain feedback on participants’ experiences with taking part in a randomised trial. A secondary aim was to examine the outcomes of the intervention itself, specifically whether participation in Mellow Babies provides improvements in the quality of mother–infant interaction post-intervention and/or improvements in mother’s wellbeing and sense of parenting self-efficacy post-intervention.

## 2. Materials and Methods

### 2.1. Design

We chose a prospective randomised open-label, blinded end-point (PROBE) clinical trial with a waiting-list control group. Outcome measures were taken at two time points: baseline (pre-intervention) and 20 weeks (post-intervention). We also sought permission to obtain follow-up information through routine health data records (such as child health screening contacts). The trial was registered with Clinicaltrials.gov: NCT02277301.

### 2.2. Participants

Women with infants less than 13 months of age who were experiencing substantial problems in developing a good relationship with their baby were eligible to take part. Participants would be expected to have significant health/social care needs including, but not limited to, mental health difficulties, problem drug use, learning difficulties, forensic issues, or children already looked after on a statutory basis. We aimed to recruit at least 50 women with infants under 13 months old. Retention rates in Mellow Babies groups are around 70% [23]. We therefore expected to have approximately 35 women with evaluable data. This would give adequate power to detect a moderate effect size on the principal outcome measure with reasonable likelihood [26].

### 2.3. Intervention

Mellow Babies is an attachment-based group parenting support programme that has been used widely throughout the U.K. and internationally. It has been used with women experiencing postnatal depression and other social and psychological difficulties [22] and retention figures are high, even among those facing the greatest adversity. Although there are CBT-based elements within MB which aim to improve anxiety and depressive symptoms, addressing parental psychological problems may be insufficient to improve parent–child relationships [27], so MB aims to enhance close parent–infant attunement directly using a combination of video feedback and hands-on practice in baby massage, interaction coaching, and infant-focused speech. Video material of mealtime interactions is shared, and mothers are encouraged to discuss solutions to parenting difficulties. The Mellow Babies programme involves attendance for 14 consecutive weeks for about 5 h per session, within usual school hours, and there is a reunion 1–3 months later. Groups can be offered on weekends, and transport, meals, and a crèche are provided.

Usual care in the present study included standard support from health and social care staff, including health visitor routine contacts and social work involvement where applicable. This was not formally recorded and would have varied across the sample but was likely to have been similar in both the intervention and waiting-list control group.

### 2.4. Mother–Infant Interaction

The primary prespecified outcome measure was direct observation of the mother–child relationship using the Mellow Parenting Observation Scale [22,28], a systematic observation of a 15 min mealtime interaction. The MPOS is an event-sampled observational system in which the frequency of positive and negative parenting behaviours is recorded and rates per minute calculated. MPOS is focused specifically on the interaction quality observed, rather than on separate behaviours of the parent or child. The scenario recorded is a normal care-taking routine, which involves the parent having to negotiate an agenda (e.g., a nappy/diaper change or mealtime), which is more reflective of a normal day-to-day parenting interaction than child-led play. Positive and negative elements are coded separately and have been shown to be statistically independent [29]. MPOS has also shown promise as a tool (using overall positive/negative interactions) for predicting future psychopathology. Previous research using banked video data of one-year-olds interacting with their parent/caregiver has shown that each increase in one positive parental behaviour per minute of observed interaction predicted 15% lower odds of a child later (age 7 years) receiving a disruptive behaviour disorder diagnosis [28]. Our previous analysis of video-recorded interactions has demonstrated moderate reliability, with an interclass correlation of 53% for positive behaviours and a Kendall’s τ of 0.6 for (non-normally distributed) negative behaviours [22]. We also showed there to be reasonable agreement between different coders for overall positive and negative behaviours, although agreement within the separate categories was less good [28].

### 2.5. Maternal Wellbeing

The Adult Wellbeing Scale (AWS) [30] is part of the U.K. Department of Health Framework for the Assessment of Children in Need and their Families, and it has been used in a number of evaluations (e.g., Wilson, et al., 2012 [31]). It is an 18-item questionnaire using 4-point Likert-type responses. Scores are generated for 4 subscales: depression, anxiety, and inwardly and outwardly directed irritability, each with a clinical cut-off (6, 8, 7, and 6, respectively). The depression and anxiety subscales are identical to those used in the Hospital Anxiety and Depression Scale [32].

The Edinburgh Postnatal Depression Scale (EPDS) [33] is a standard clinical screening tool for postnatal depression that is widely used by health visitors. It is a 10-item questionnaire using Likert-type responses. The maximum possible score is 30, and a cut-off score of 14 or more is taken to indicate depressive illness, with anything over 10 indicating possible depression.

### 2.6. Background Information and Interview Feedback

We devised a brief questionnaire for obtaining demographic information, data on family background, and reason for referral to the group (baseline only). We also developed a practitioner interview topic guide asking about their role, experience with delivering the Mellow Babies Programme, the challenges with delivering the programme, what worked well, and what did not work well. All practitioners were invited to interview, with a view to ceasing participant interviews once saturation (i.e., no new themes arising) was achieved.

### 2.7. Setting and Procedure

This study was conducted in three areas of Northern Ireland: Lisburn, North Down and Ards, and Downpatrick, and recruitment took place between September 2014 and December 2015. Eligible participants were identified by a practitioner in the Child and Family team, which included social work, mental health, and visiting health (public health nurse) practitioners. The Mellow Babies practitioner asked participants if they were happy for a university-based research nurse to contact them to hear more about this study. The research nurse was trained in appropriate procedures (including Good Clinical Practices training) and was experienced in working with research participants with a range of health and social care needs. Consenting women were given information on this study, specifically being informed that participation would involve either receiving the intervention the following month or waiting for the next group to begin, after a delay of approximately 20 weeks. Signed consent was sought by the research nurse after an interval of at least 24 h to allow the women to arrive at an informed decision. Women agreeing to participate were asked to complete baseline measures, including a 15 min video of a mealtime (recorded by the Mellow Babies practitioner after the mother had become comfortable with her presence in the home). This is the standard setting for a video made at the beginning of all Mellow Babies groups, and the use of a caregiving scenario is ideal for using the Mellow Parenting Observation System (see below). Videos were burned onto two DVDs, one to give to the mother and another to be sent by secure mail with a study identifier number only to the university research office. Videos were assessed by an independent coder blind to group allocation and trained to research reliability level in MPOS. Following completion of the baseline measures, the Child and Family team member telephoned the university research office to obtain a study code and a group allocation. The study code was transferred to the paperwork and later to the burned DVD. At the end of the intervention/waiting-list period, all the baseline measures were repeated for the intervention and control groups. Finally, semistructured interviews were carried out by the research nurse with a sample of mothers and practitioners working in two different regions. Practitioners and mothers were invited to give feedback during one-to-one phone interviews lasting approximately 30 min. Feedback from practitioners on the experience of running groups within a research trial is reported here.

This study was approved by the North of Scotland Research Ethics Service, reference 14/NS/1007.

### 2.8. Analysis

Differences in all quantitative measures between the intervention and control conditions at the end of the study period were assessed using one-way between-subjects analysis of covariance (ANCOVA), accounting for baseline scores, using IBM SPSS Statistics 24 software. Checks were carried out to ensure homogeneity of regression and linear relationship between the covariate (baseline ratings) and dependent variable (post-intervention ratings). Feedback interviews were digitally recorded following informed consent, transcribed verbatim, coded, and analysed descriptively using thematic analysis [34,35].

## 3. Results

### 3.1. Feasibility

This study eventually ran over two years and three rounds of interventions (whilst it had originally been planned for one year and one round of intervention). With reference to Figure 1, of the 45 women referred to the research, 7 were ineligible to take part. The remaining 38 were eligible and gave consent to take part in the study. Of these, 1 did not complete baseline (T1) research measures, 7 completed baseline measures but dropped out before post-assessment (T2), and 28 (74% of those randomised; 15 intervention/13 control) completed the study and returned both baseline and post-assessment questionnaires, 21 of whom returned all questionnaires and video data at both time points. A further two returned videos only (no questionnaires). Table 1 below summarises the demographic characteristics of the participants at baseline. There were no statistically significant differences between the two randomisation arms.

### 3.2. Parent–Infant Interaction

Twenty-three participants returned both baseline and post-intervention videos. The baseline positive interaction rating was significantly related to the post-intervention positive rating: *F*(1,22) = 17.627; *p* < 0.001; partial η^2^ = 0.45. Adjusting for this covariate resulted in a statistically significant effect of the between-subjects factor group (intervention/control): *F*(1,22) = 6.434; *p* = 0.019; partial η^2^ = 0.23. The adjusted mean score for the intervention group was 0.111 compared to 0.088 for the waiting-list group. There was no significant change in negative parenting behaviours, but few negative behaviours were observed.

### 3.3. Maternal Wellbeing

There were no significant differences from pre- to post-intervention for either the intervention or control groups on any of the questionnaires (Table 2).

### 3.4. Interview Feedback

The practitioner interviews highlighted significant challenges in recruiting mothers to the trial. One problematic feature was the need to recruit twice as many mothers to populate two arms of the study (as opposed to the usual practice of only recruiting enough for the intervention group). However, the most difficult aspect of recruitment was that most of the mothers did not want to be randomised into the (waiting-list) control group; they were unwilling to wait another four months before receiving the Mellow Babies programme. Some mothers declined to be part of the research because of this, and some withdrew when they were randomised to the waiting-list arm.

## 4. Discussion

The feasibility pilot study showed that conducting a larger trial of Mellow Babies (vs. no intervention) is feasible, providing learning from this study is taken into account. We achieved good recruitment and reasonable retention rates, positive feedback from practitioners, as well as useful feedback on challenges and barriers to participation. There was an indication of improvement in the quality of mother–infant interactions, although the sample is too small to draw firm conclusions. No statistically significant difference in either the frequency of negative interaction behaviours or in maternal wellbeing scores was observed.

The application of these findings to the design of a larger RCT is limited due to the use of a waiting list design, meaning that all participants received the intervention eventually. This made long-term follow-up, essential in a definitive trial, of little value, and it is possible that natural improvement among those mothers allocated to the waiting-list condition was curtailed until they joined a group [36]. The reasons for using a waiting0list design were both practical/financial and ethical. The disadvantage to this approach is that, as the Mellow Babies programme was already being offered in the area with no alternative intervention available, not to offer the intervention to all participants would be considered as withholding access to usual care. This is the nature of research of this type: the need to generate better-quality evidence about already established interventions leads to debate as to what constitutes appropriate ‘usual care’ and whether withholding of a potentially promising intervention could be considered ethically unsound.

The positive change in parent–infant interaction scores is encouraging and in keeping with earlier research on Mellow Babies [22]. However, both the present study and the previous RCT were based on relatively small samples with complete data (17 and 28, respectively) and both employed waiting-list designs. Recent reviews and meta-analyses were not able to examine the impact on parent–infant interaction as too few of the included studies collected observation data [23,37]. It was difficult to assess the impact on negative interactive behaviours as these tended not to occur with sufficient frequency to allow assessment. Whether a sustained positive impact on parent–infant interaction can be attributed to Mellow Babies will require a larger RCT with long-term follow-up and consideration of the impact of the research process on the objective observation of behaviour. Embedding such an RCT within a comprehensive and pragmatic evaluation context such as that outlined in the MRC Complex Interventions Model will be essential [38].

A previous meta-analysis has been able to demonstrate some impact on maternal and child mental health outcomes, although findings have been mixed and subject to methodological bias [37]. We found no impact on maternal wellbeing measured immediately post-intervention, which was surprising given previous findings. During a dissemination event in the area where this study was conducted, there was considerable discussion regarding a local culture of mistrust, which could mean that participants were highly likely to understate any mental wellbeing concerns at baseline, whereas by the post-intervention stage, following several weeks of an intervention with trust-building and nurturing explicitly built in, responses were likely to have been more truthful. This can only be considered speculation at this stage.

Whilst the present study has provided useful information regarding the practical reality of conducting a definitive trial of the Mellow Babies intervention, there are some methodological limitations which need to be highlighted. The reliance on the existing services context and concomitant reliance on a waiting-list control design have been discussed above. In addition, the main inclusion criterion was that mothers had ‘substantial problems in developing a bond with their baby’, but this factor was not measured directly (rather, referral was made according to proxy criteria and practitioner professional opinion). The lack of long-term follow-up means we could not assess the feasibility and likely retention to final follow up. Finally, this was a mother-only sample. Mellow Babies is designed to be conducted in single-sex groups, and there is a version adapted for fathers. Whether they would readily engage in a research trial has not been assessed.

## 5. Conclusions

Conducting a randomised trial of Mellow Babies is feasible and acceptable, but there remain significant difficulties in recruiting critical numbers from this population, which need to be addressed in the design of any future trial. Participants in Mellow Babies showed significant improvement in the quality of their interaction with their babies, but no improvement in self-reported maternal mental wellbeing.

## Figures and Tables

**Figure 1 children-11-00510-f001:**
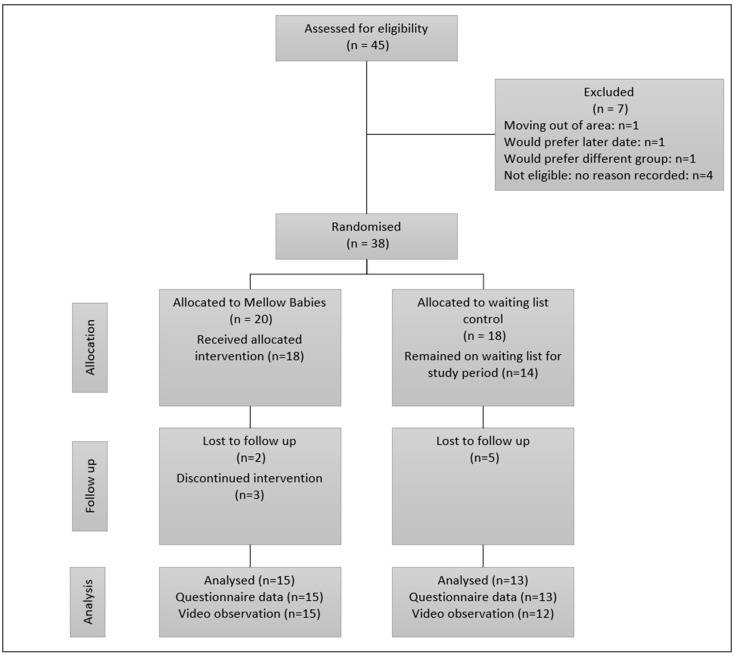
CONSORT diagram.

**Table 1 children-11-00510-t001:** Sample characteristics.

	All (*n* = 38)	Intervention (*n* = 20)	Control (*n* = 18)	*p*
Age of mother, years; mean (SD)	24.0 (5.6)	23.5 (4.9)	24.5 (6.5)	0.6
Sex of baby; *n* (%)				
Female	15 (39.5)	7 (35.0)	8 (44.4)	0.4
Male	19 (50.0)	11 (55.0)	8 (44.4)	
Missing	4 (10.5)	2 (10.0)	2 (11.1)	
Age of baby, months; mean (SD)	5.44 (4.44)	4.39 (3.50)	6.63 (5.4)	0.2
Number of other children at home; *n* (%)				
0	30 (78.9)	17 (85.0)	13 (72.2)	0.8
1	2 (5.3)	0	2 (11.1)	
2	1 (2.6)	1 (5.0)	0	
3	1 (2.6)	0	1 (5.6)	
Missing	4 (10.5)	2 (10.0)	2 (11.1)	
Reason for referral; *n* (%)				
Mental health	17 (44.7)	10 (50.0)	7 (38.9)	0.4
Leaving care	6 (15.8)	4 (20.0)	2 (11.1)	0.6
Child protection	16 (42.1)	8 (40.0)	8 (44.4)	1.0
Homeless	2 (5.3)	12 (60.0)	2 (11.1)	0.5
Domestic violence	5 (13.2)	3 (15.0)	2 (11.1)	1.0
Complex social needs	12 (31.6)	6 (30.0)	6 (33.3)	1.0
Criminal justice	5 (13.2)	3 (15.0)	2 (11.1)	1.0
Substance misuse	7 (18.4)	4 (20.0)	3 (16.7)	1.0
Missing	14 (36.8)	8 (40.0)	6 (33.3)	
Reasons for referral—number endorsed; *n* (%)				
0	1 (2.6)	0	1 (5.6)	0.5
1	5 (13.2)	3 (15.0)	2 (11.1)	
2	5 (13.2)	1 (5.0)	4 (22.2)	
3	4 (10.5)	2 (10.0)	2 (11.1)	
4	4 (10.5)	3 (15.0)	1 (5.6)	
5	4 (10.5)	3 (15.0)	1 (5.6)	
6	0	0	0	
7	1 (2.6)	0	1 (5.6)	
Missing	14 (36.8)	8 (40.0)	6 (33.3)	

**Table 2 children-11-00510-t002:** Descriptive statistics and analysis outcomes for primary and secondary measures.

		Baseline		Post		Between-Subjects Factor		
		Mean (SD)		Mean (SD)				Partial
Measure	Subscale	MB	WL	MB	WL	*F*	*p*	η^2^
MPOS	Positive	0.081 (0.042)	0.099 (0.048)	0.107 (0.031)	0.092 (0.025)	6.434	0.019	0.226
	Negative	0.002 (0.005)	0.005 (0.011)	0.004 (0.006)	0.011 (0.015)	1.827	0.190	0.077
EPDS	Total	10.42 (5.98)	11.67 (6.80)	9.98 (6.46)	14.00 (7.35)	0.113	0.741	0.006
AWBS	Depression	5.50 (3.06)	5.44 (2.79)	5.50 (3.21)	5.33 (2.74)	0.137	0.714	0.006
	Anxiety	5.42 (3.63)	6.89 (3.92)	7.58 (3.09)	7.22 (4.09)	0.751	0.395	0.030
	Irritability-I	2.25 (2.01)	2.33 (3.57)	3.17 (2.73)	2.44 (3.01)	0.980	0.332	0.038
	Irritability-O	2.67 (1.97)	4.33 (2.65)	3.67 (2.67)	4.22 (3.67)	0.952	0.339	0.037

MPOS: Mellow Parenting Observation Scale; EPDS: Edinburgh Postnatal Depression Scale; AWBS: Adult Wellbeing Scale; Irritability-I: inwardly directed irritability; Irritability-O: outwardly directed irritability. Bold text indicates statistically significant difference.

## Data Availability

The data presented in this study are available on request from the corresponding author due to limited resources precluding placement in a public repository.

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
