# Peer review of "Mellow Babies: A Randomised Feasibility Trial of an Intervention to Improve the Quality of Parent–Infant Interactions and Parental Mental Wellbeing"

_children, 2024, doi:10.3390/children11050510_

Round 1

Reviewer 1 Report

Comments and Suggestions for Authors

This intervention study with mellow babies and mothers attracted my attention. This program seems to be a useful initiative that helps prepare mothers for motherhood and positively affects the health of their babies. However, although there is a serious effort in your study, some deficiencies make it confusing for the publication of the study. Writing the method section in a certain order was deemed successful. However, I could not find detailed information about the scales you used. Who developed the scales? Has its validity and reliability been checked? What are the analyzes performed for validity and reliability? What did you find about Cronbach Alpa in this study? In particular, there is no information about the Mellow Parenting Observation Scale in your study. It seems that you made an evaluation using only this scale. Who developed this scale? What are the measurement criteria of this scale? How is it used? What is the evidence that it is valid and reliable? There are no such things. This being the case, many questions arise regarding the reliability of the study. Additionally, a more detailed explanation is needed about the Mellow Babies program you used in this study. Yes, you explained the application a little bit. But how did you do this application? In what environment did you plan your education? What kind of applications did you provide training on? How did the mothers feel? Has it improved? What were the mothers' knowledge levels about their babies before the training, and were there any differences between the groups? What were the circumstances that would affect the effective implementation of this program? For example, does the age of the mothers have an impact on this program, as well as the mother's number of children. I think such data are factors that will affect the study. Additionally, there is no information about the homogeneity of the two groups. This is a very important criterion for interventional studies. Also, the sampling inclusion and exclusion criteria are not written. Can every mother participate in this program? There are many questions that need to be resolved. Additionally, your sample number is very small, which negatively affects the study. I think the program you use is a routinely applied program. Why didn't you try to develop a program on this subject?

Author Response

Response to reviewer 1

Reviewer comment

Our response

Line number/s

This intervention study with mellow babies and mothers attracted my attention. This program seems to be a useful initiative that helps prepare mothers for motherhood and positively affects the health of their babies. However, although there is a serious effort in your study, some deficiencies make it confusing for the publication of the study.

None required, but please see our responses below.

Writing the method section in a certain order was deemed successful. However, I could not find detailed information about the scales you used. Who developed the scales? Has its validity and reliability been checked? What are the analyzes performed for validity and reliability? What did you find about Cronbach Alpa in this study? In particular, there is no information about the Mellow Parenting Observation Scale in your study. It seems that you made an evaluation using only this scale. Who developed this scale? What are the measurement criteria of this scale? How is it used? What is the evidence that it is valid and reliable? There are no such things. This being the case, many questions arise regarding the reliability of the study.

We have provided references for each of the standard measures to allow the reader to access full information on their validity and reliability. We believe adding all this detail to the manuscript would have made it too lengthy. This is an appropriately parsimonious approach given that both the EPDS and AWBS are standard measures in the field and will be well-recognised by readers.

With regard to the MPOS, developed by Puckering (see citations 22 and 28), we have stated how it was used in the study, but have added some more information to the paper on reliability.

“Our previous analysis of video-recorded interactions has demonstrated moderate reli-ability, with an inter-class correlation of 53% for positive behaviours and a Kendall’s τ of 0.6 for (non-normally distributed) negative behaviours [22].” We have also stated that the MPOS was pre-specified as our primary outcome.

150-153

Additionally, a more detailed explanation is needed about the Mellow Babies program you used in this study. Yes, you explained the application a little bit. But how did you do this application? In what environment did you plan your education? What kind of applications did you provide training on?

As above, we provide a reference to a paper which describes the intervention in more detail. We believe the information provided in the ‘Intervention’ section is sufficient for the reader to understand the paper. We have however added some wording to line 120:

“…weeks for about 5 hours per session,..”

109-126

How did the mothers feel? Has it improved? What were the mothers' knowledge levels about their babies before the training, and were there any differences between the groups? What were the circumstances that would affect the effective implementation of this program? For example, does the age of the mothers have an impact on this program, as well as the mother's number of children. I think such data are factors that will affect the study.

Mothers’ feelings were measured by both the AWBS and the EPDS. The outcome of this analysis is already included in the paper.

We did not assess mothers’ knowledge about their babies as increasing knowledge is not a primary aim of the intervention.

There are a range of circumstances which could impact the intervention, and we have discussed these as relevant to this study in the Discussion section. Ultimately, this intervention is designed to overcome most circumstantial factors and to be flexible around the participants’ needs, inasmuch as is practical. A full explication of all circumstances which may impact the intervention is not possible in this paper which was a feasibility study for a definitive randomised trial. A future definitive trial would of course involve a realist examination of key questions about for whom the interventions works and in what circumstances – but this could not be addressed within this study design.

Table 2

Additionally, there is no information about the homogeneity of the two groups. This is a very important criterion for interventional studies.

We agree that group homogeneity is an important issue in group-based intervention studies, and future work will address this issue. No significant heterogeneity within the groups was noted by the practitioners, and for a sample of this size it was not deemed appropriate to describe the participants in any greater detail than we already have. We have provided baseline characteristics for both groups, including statistical analyses on whether they differed significantly from each other.

Table 1

Also, the sampling inclusion and exclusion criteria are not written. Can every mother participate in this program?

The inclusion criteria are stated in the paper thus:

“Women with infants less than 13 months of age who were experiencing substantial problems in developing a good relationship with their baby were eligible to take part. Participants would be expected to have significant health / social care needs, including but not limited to mental health difficulties, problem drug use, learning difficulties, forensic issues, or they may have children already looked after on a statutory basis.” So no, not every mother could participate. Eligibility was left to the discretion of the recruiting practitioner, who noted the reasons for eligibility (reported in results).

100-104

Table 1

There are many questions that need to be resolved. Additionally, your sample number is very small, which negatively affects the study.

We acknowledge the small sample in the Discussion section. However, it should be borne in mind that this was a feasibility trial and we recruited more than expected (our target was 35).

279-281

106

I think the program you use is a routinely applied program. Why didn't you try to develop a program on this subject?

We are not sure what the reviewer means with this comment. We did not try to develop a new programme as there is no need for a new programme. Our aim was to investigate the feasibility of conducting a large RCT of an already-established programme to allow its evidence base to be improved.

Reviewer 2 Report

Comments and Suggestions for Authors

Thank you for sharing your work.

Please explain more about the Child and Family team - is this for Infant mental health, or developmental disability. This would help the international audience.

Four were not eligible for 'other reasons' which need to be described.

Just to make it clear, the Mellow Babies practitioner was present during the video recording of the mealtime? We have trialled something similar and there are cultural challenges to consider in having an unfamiliar person in the house at a mealtime who is not eating with the family.

This is a fascinating project. Ultimately I wonder whether an RCT is the right aim for the work. I would be interested for the authors to consider alternative study design that might take a Realist or pragmatic view (RAMESES). The conclusion that a trial is feasible but challenged does not seem to make the most of the experience and I look forward to reading alternative ideas, given that your experience is probably very relatable amongst non-drug interventions.

Author Response

Response to reviewer 2

Reviewer comment

Our response

Line number/s

Thank you for sharing your work.

Thank you for reviewing it!

Please explain more about the Child and Family team - is this for Infant mental health, or developmental disability. This would help the international audience.

This is a multidisciplinary team where families with any additional support need could be referred. Referring practitioners included social work, mental health, and health visiting practitioners. We have added wording to clarify this.

“Eligible participants were identified by a practitioner in the Child and Family team, which included social work, mental health, and health visiting (public health nurse) practitioners.”

180-182

Four were not eligible for 'other reasons' which need to be described.

This information represents missing data. In most cases, practitioners indicated if a potential participant was not eligible, and then provided an explanation. In the case of these four, the term ‘not eligible’ was given, with no detail. We have changed the language on the CONSORT diagram to ‘not eligible – no reason recorded’ to reflect this.

Figure 1

Just to make it clear, the Mellow Babies practitioner was present during the video recording of the mealtime? We have trialled something similar and there are cultural challenges to consider in having an unfamiliar person in the house at a mealtime who is not eating with the family.

Thank you, this is an important point. The practitioner was indeed present while carrying out the video recording. The Mellow Babies model includes practitioners taking time to sensitively build rapport with the individual group participants prior to the intervention commencing. The recording of the video would never have taken place on an initial home visit, and practitioners are trained and experienced in ensuring interactions recorded are as natural as possible. However, we acknowledge this will not work in every cultural context, and that future studies will need to consider this.

We have added wording to reflect this:

“…(recorded by the Mellow Babies practitioner after the mother had become comfortable with her presence in the home).”

192-193

This is a fascinating project. Ultimately I wonder whether an RCT is the right aim for the work. I would be interested for the authors to consider alternative study design that might take a Realist or pragmatic view (RAMESES). The conclusion that a trial is feasible but challenged does not seem to make the most of the experience and I look forward to reading alternative ideas, given that your experience is probably very relatable amongst non-drug interventions.

Yes – we agree that more creative approaches are needed for complex intervention projects such as these. We endeavoured to follow the MRC Complex Interventions Framework (https://www.bmj.com/content/374/bmj.n2061) in designing the larger trial, and any RCT of such an intervention should be embedded in a broader process such as that outlined in this model and should adhere to the MRC process evaluation guidelines.  We have added wording to the Discussion:

“Embedding such an RCT within a comprehensive and pragmatic evaluation context such as that outlined in the MRC Complex Interventions Model will be essential [38]”

287-289

Round 2

Reviewer 1 Report

Comments and Suggestions for Authors

Study is ok. Athours edit study enough. It can be publish.

Author Response

Thank you for approving our paper for publication.

Reviewer 2 Report

Comments and Suggestions for Authors

The suggested revisions have been responded to appropriately and the manuscript is of value to the community.

Author Response

(The authors gave the same response as above.)
